# X-ray Radiotherapy Impacts Cardiac Dysfunction by Modulating the Sympathetic Nervous System and Calcium Transients

**DOI:** 10.3390/ijms25179483

**Published:** 2024-08-31

**Authors:** Justyne Feat-Vetel, Nadine Suffee, Florence Bachelot, Morgane Dos Santos, Nathalie Mougenot, Elise Delage, Florian Saliou, Sabrina Martin, Isabelle Brunet, Pierre Sicard, Virginie Monceau

**Affiliations:** 1Laboratoire de Pharmacologie Expérimentale et Moléculaire (LPEM), Service d’Ingénierie Moléculaire Pour la Santé (SIMoS), Département Médicaments et Technologies Pour la Santé (DMTS), CEA, 91191 Gif-sur-Yvette, France; justyne.vetel@cea.fr; 2UMR 1166, Unité de Recherche sur les Maladies Cardiovasculaires et Métaboliques, INSERM, 75013 Paris, France; nadine.suffee@inserm.fr; 3PSE-SANTE/SESANE/LRTOX, Institut de Radioprotection et de Sûreté Nucléaire-IRSN, 92260 Fontenay-aux-Roses, France; florence.bachelot@irsn.fr (F.B.); florian.saliou@irsn.fr (F.S.); 4PSE-SANTE/SERAMED/LRAcc, Institut de Radioprotection et de Sûreté Nucléaire-IRSN, 92260 Fontenay-aux-Roses, France; morgane.dossantos@irsn.fr; 5UMS28, INSERM, Sorbonne Université, Plateforme PECMV, 75005 Paris, France; nathalie.mougenot@inserm.fr; 6CellTechs Laboratory, SupBiotech, 94800 Villejuif, France; elise.delage@supbiotech.fr; 7Service d’Etude des Prions et des Infections Atypiques, Institut François Jacob, Commissariat à l’Energie Atomique et aux Energies Alternatives, Université Paris Saclay, 91405 Fontenay-aux-Roses, France; 8Center for Interdisciplinary Research in Biology (CIRB), College de France, 75001 Paris, France; sabrina.martin@college-de-france.fr (S.M.); isabelle.brunet@college-de-france.fr (I.B.); 9PhyMedExp, IPAM/Biocampus, INSERM, CNRS, Université de Montpellier, 34095 Montpellier, France; pierre.sicard@inserm.fr

**Keywords:** breast cancer, cardiovascular disease, radiotherapy, ionizing radiation

## Abstract

Recent epidemiological studies have shown that patients with right-sided breast cancer (RBC) treated with X-ray irradiation (IR) are more susceptible to developing cardiovascular diseases, such as arrhythmias, atrial fibrillation, and conduction disturbances after radiotherapy (RT). Our aim was to investigate the mechanisms induced by low to moderate doses of IR and to evaluate changes in the cardiac sympathetic nervous system (CSNS), atrial remodeling, and calcium homeostasis involved in cardiac rhythm. To mimic the RT of the RBC, female C57Bl/6J mice were exposed to X-ray doses ranging from 0.25 to 2 Gy targeting 40% of the top of the heart. At 60 weeks after RI, Doppler ultrasound showed a significant reduction in myocardial strain, ejection fraction, and atrial function, with a significant accumulation of fibrosis in the epicardial layer and apoptosis at 0.5 mGy. Calcium transient protein expression levels, such as RYR2, NAK, Kir2.1, and SERCA2a, increased in the atrium only at 0.5 Gy and 2 Gy at 24 h, and persisted over time. Interestingly, 3D imaging of the cleaned hearts showed an early reduction of CSNS spines and dendrites in the ventricles and a late reorientation of nerve fibers, combined with a decrease in SEMA3a expression levels. Our results showed that local heart IR from 0.25 Gy induced late cardiac and atrial dysfunction and fibrosis development. After IR, ventricular CSNS and calcium transient protein expression levels were rearranged, which affected cardiac contractility. The results are very promising in terms of identifying pro-arrhythmic mechanisms and preventing arrhythmias during RT treatment in patients with RBC.

## 1. Introduction

Epidemiological studies show long-term side effects of RT on the development of cardiovascular diseases (CD), including coronary artery disease, pericarditis, cardiomyopathies, valvular heart disease, and conduction abnormalities leading to rhythm disorders [1,2]. The development of these cardiomyopathies appears to be strongly dependent on both the dose and the volume of heart exposed to X-ray IR [3,4]. While radiation-induced cardiac dysfunction is mostly asymptomatic, and heart failure is rare during the first five years after irradiation, many studies of cardiac function show relatively acute (>one year) changes in systolic and diastolic function, myocardial perfusion, dysrhythmias, and conduction defects [5,6,7].

In particular, conduction abnormalities and arrhythmias were the clinical manifestations most frequently observed in post-IR patients. Indeed, Bansod et al. showed, in a non-trivial study, that 2.3% of hospitalized breast cancer survivors developed arrhythmic death over a seven-year period when they received RT during treatment [8]. Interestingly, patients who did not receive RT during their treatment did not develop arrhythmias or conduction disturbances [9], and the risk of developing these rhythm disturbances is higher in RBC patients [7,10]. In addition, recent studies have shown that chronic atrial arrhythmias can occur in patients following RT [11]. According to the 2020–2023 guideline, the management recommendation for the diagnosis of atrial fibrillation (AF) in cancer patients is now at the time of treatment, in particular by targeted electrical cardioversion [12,13]. Our previous work has also shown an association between the development of arrythmias and RT treatments in patients with RBC [14]. In fact, major areas of the heart are most exposed during RT for RBC, in particular, the CSNS, the atrio-ventricular (AV) and sinus-atrial (SA) cardiac nodes, and the right atrium. In addition, the conduction pathways are key players in this cardiac remodeling [15,16]. Arrhythmia is also an irregular heart rhythm, suggesting that the CSNS plays an important role in vascular remodeling and in the physiology and maintenance of cardiac function [17,18]. In our studies, based on the patients’ definitive treatment plans, dosimetry was performed at the end of treatment on the different cardiac regions in the patients’ scans. The dosimetric results showed a cumulative dose range for the atria, AV, and SA ganglia of between 0.4 and 1.6 Gy when the heart was exposed for the right breast cancer [19]. This pathological state of arrhythmia may be caused by several factors, but one of the most important is the disruption of calcium (Ca^2+^) homeostasis [20,21]. The Ca^2+^ homeostasis and its effect on contractility is an essential element in any excitation–contraction coupling event.

Our hypothesis is that RT-induced changes in the CSNS, as well as atrial remodeling and Ca^2+^ homeostasis, are involved in the development of cardiac arrhythmias. However, no study has investigated the effect of low to moderate doses on the development of cardiac pathologies by targeting the CSNS. Short-term radiation-induced iatrogenic effects can be explained by the cytotoxic mode of action of ionizing radiation. However, the mechanisms involved in cardiac toxicity, particularly long-term arrhythmias, are more complex and, for these reasons, represent a public health problem that is still poorly understood. In this study, we used a mouse model of localized irradiation in the context of single-dose exposure with the same final cumulative dose ranging from 0.25 Gy to 2 Gy. To this end, we investigated the mechanisms involved in the development of radiation-induced arrhythmias to provide new strategies for the prevention of arrhythmias during RT treatment in RBC patients.

## 2. Results

### 2.1. Effects of Top-Heart X-ray Irradiation on Atrial and Ventricular Cardiac Function in Mice 60 Weeks Post-IR

Animal weights were stable throughout the follow-up period of irradiation, and no animals showed signs of cardiac toxicity or died unexpectedly. Because phosphorylation of H2AX at Ser 139 (γ-H2AX) is abundant, rapid, and well correlated with each double-strand break, it is the most sensitive marker that can be used to examine the DNA damage produced and the subsequent repair of that damage. We used it in immunolabeling to show the location of irradiation and to validate our model of irradiation at the top of the heart (Appendix A).

At 60 weeks post-irradiation, the cardiac function evaluated revealed a significant decrease of the left ventricular ejection fraction (LVEF) in 0.25 Gy to 2 Gy irradiated mice compared to NIR mice (Figure 1(Aa)). Similarly, the diastolic left ventricular wall thickness decreased significantly, indicating myocardial remodeling, in 0.25 Gy to 2 Gy irradiated mice compared to NIR mice (Figure 1(Ab)). These results suggest that top-heart X-ray irradiated at 60 weeks post-IR induced an impaired cardiac function (Appendix A).

Using the VevoStrain software for LV deformation analysis, we showed a significant increase in dyssynchrony in 2 Gy irradiated mice, as compared to NIR mice (Figure 1(Ba,Bb)), and a significant decrease in strain in mid and apical segments in 2 Gy mice, as compared to NIR mice (Figure 1(Bc–Be)). These results suggest that the top-of-heart X-ray IR induced an increase in myocardial dyssynchrony associated with incorrect myocardial deformation and function (reduction in the ejection fraction) at 60 weeks post-IR.

In the basal electrocardiogram shown in all experimental groups, the QRS complex with interval duration was significantly shortened compared to NIR mice (Figure 1(Ca)), indicating that ventricular conduction was largely affected by IR. Subsequently, a significant decrease in RR interval duration was observed in all irradiated mice, as compared to NIR mice, which correlates with the significant increase in heart rate at 2 Gy compared to NIR mice (Figure 1(Cb,Cc)). In addition, at the level of the atria, P-wave duration was reduced in all irradiated mice compared to NIR mice (Figure 1(Cd)). There was also a significant increase in the rate of deformation, indicating increased ventricular relaxation in all irradiated mice. This is also consistent with diastolic dysfunction (Figure 1(Bf)). After bust pacing of the atria, all irradiated mice showed a reduction in AF duration (Figure 1D), suggesting a loss of atrial function after IR, regardless of the dose exposed to the atria. In addition, we observed a significant increase in diastolic volume at 0.25 Gy, which may lead to poor filling of the left atrium and contribute to diastolic dysfunction of the left ventricle (Figure 1E). All these results suggest that top-heart X-ray IR induced a decrease in general atrial activities and a reduction in ventricular depolarization at 60 weeks post-IR, resulting in impaired transmission of myocardial electrical activity.

### 2.2. Effect of Top-Heart X-ray Irradiation on Atrial Morphology and Remodeling

We have previously shown an alteration in atrial and ventricular cardiac function, as well as an increase in atrial volume, following upper cardiac X-ray IR 60 weeks after IR. To complement these findings, we assessed cardiac morphology at an early stage, 24 h post-IR, and at a late stage, 60 weeks post-IR.

The dysfunction of the atria suggests cell death of the tissue after irradiation. To this end, cell death was examined by the expression of cleaved caspase-3 protein at 24 h post-IR and 60 weeks post-IR. The samples were divided into two parts: the upper part of the heart with atria + cardiac nodes and the lower part of the heart with ventricles. Twenty-four hours after IR, the expression of cleaved caspase-3 protein was significantly increased in the atria of 2 Gy irradiated mice compared to NIR mice, whereas no effect was observed in the ventricles (Figure 2(Aa,Ab)). At 60 weeks post-IR, we obtained similar results at 0.5 Gy and 2 Gy (Figure 2(Ba,Bb)). These data suggest that X-ray IR of the upper part of the heart induces cell death only in the part of the heart affected by the IR, i.e., the upper part of the heart containing the atria and cardiac nodes.

Tissue remodeling was assessed by Sirius Red staining at 60 weeks post-IR, which showed a slight significant increase in interstitial cardiac fibrosis in 0.5 and 2 Gy irradiated mice compared to NIR mice (Figure 2(Ca)). The results also showed that epicardial fibrosis deposition was significantly higher in the atria than in the venules (Figure 2(Cc,Cd)) in 0.5 Gy and 2 Gy irradiated mice compared to NIR mice. These data provide evidence for tissue remodeling and relaxation in the upper part of the heart 60 weeks after IR.

In order to understand the effect of atrial and ventricular dysfunction, we investigated the role of the CSNS in mice irradiated to the top of the heart. Labeling with a specific marker of CSNS, tyrosine hydroxylase (TH), and using the iDISCO+ method (Figure 3(Aa)), we found no significant difference in the total number of filaments (Figure 3(Ab)) or the number of dendritic segments (Figure 3(Ac)), and no significant difference in filament volume (Figure 3(Ad)) or length (Figure 3(Ae)) at 24 h post-IR. However, a significant decrease in spine density on a dendritic segment was observed in 0.25 Gy and 0.5 Gy irradiated mice compared to NIR mice (Figure 3(Af)), which was associated with a significant decrease in the dendritic branching level in all experimental groups compared to NIR mice (Figure 3(Ag)). These results suggest that the dendritic spines of the arborization of TH-positive fibers were targeted early during irradiation at the top of the heart.

In addition, catecholamine and Ca^2+^ concentrations were measured in heart samples 24 h post-IR. Our results showed a significant increase in the catecholamine concentration in 2 Gy irradiated mice compared to NIR mice (Figure 3B). There was also a significant decrease in the Ca^2+^ concentration in cardiac tissue of 0.25 Gy to 2 Gy irradiated mice compared to NIR mice (Figure 3C). These results suggest that top-heart X-ray IR induced an early increase in the tissue catecholamine concentration, which could be the cause of cardiac dysfunction by inducing an imbalance in intracellular Ca^2+^, mainly at early stages after IR.

### 2.3. The Top-Heart X-ray Irradiation Induced a Delayed Reorganization of Cardiac SNS

At 60 weeks post-IR, 3D imaging showed a trend toward an increase in the total number of filaments in mice irradiated with 2 Gy compared to NIR mice (Figure 4a,b). However, other parameters, such as the size or number of dendrites or spines, did not change, in contrast to what we observed in hearts 24 h post-IR (Figure 4c–g). The location of TH-positive cardiac SNS fibers was analyzed within the irradiated heart (Figure 5(Aa)). The results of the quantification unexpectedly showed a significant increase in the total number of filaments (Figure 5(Ab)) and a significant increase in the level of segmental branches inside the heart of mice irradiated at 2 Gy compared with NIR mice (Figure 5), in which the nerve fibers were mainly located at the periphery. This observation was also found for the other irradiated groups, in the form of a trend. These results indicate that X-ray IR of the upper part of the heart would induce a reorganization of the TH-positive fibers of the CSNS 60 weeks post-IR. In addition, cardiac innervation is regulated by growth factors and chemo-repulsive factors, such as semaphorin 3A (SEMA3a). No difference in SEMA3a expression was observed in the upper part of the heart (Figure 5B). In contrast, in the lower part of the heart, SEMA3a was significantly decreased in hearts exposed to 0.25 Gy and 0.5 Gy (Figure 5B). SEMA3a is an axonal guidance molecule and plays a crucial role in cardiac patterning [22]. In addition, CSNS function was analyzed by measuring the catecholamine concentration in heart samples and β1-adrenergic receptor (ADRB1) protein expression. There was no significant effect on tissue catecholamine levels (Figure 5C), but there was an increase in ADRB1 in the ventricles of irradiated hearts (Figure 5D), suggesting a compensatory mechanism. Taken together, these results suggest that top-heart X-ray IR induces a reorganization of cardiac SNS fibers associated with a downregulation of SEMA3a protein expression and an increase in β1-adrenergic receptor.

### 2.4. Effect of Top-Heart X-ray Irradiation on Excitation–Contraction Coupling in Atrial Cardiomyocytes

To investigate the effect of IR on the excitation–contraction coupling (ECC) in atrial cardiomyocytes, we focused on the Ca^2+^ partners RYR2, SERCA2a, PLNp, NCX, NKA, and Kir2.1. First, at 24 h post-IR, the data showed no difference in the ventricular part and a significant increase in the protein expression of RYR2 and SERCA2a in 2 Gy irradiated mice compared to NIR mice, in the upper part of the heart containing atria and nodes (Figure 6(Aa,Ab)). The PLN pentamer/PLN monomer ratio, which determines the balance between storage and SERCA2a inhibition, was decreased in irradiated mice only in the upper part of the heart (Figure 6(Ac)). The protein expression level of NCX, which is the main cardiac extrusion mechanism for the Ca^2+^ for cardiomyocyte relaxation, was not changed 24 h after irradiation in the upper and lower parts of the heart compared to NIR mice (Figure 6(Ad)). The expression of the NKA pump was significantly increased in 2 Gy irradiated mice compared to NIR mice only in the upper part of the heart (Figure 6(Ae)). The expression level of Kir2.1, which is an important component of cardiac terminal repolarization and resting membrane stability, was significantly increased in the 0.25 Gy and 2 Gy irradiated mice in the upper part of the heart, whereas in the ventricles, the amount of protein was significantly decreased in 2 Gy irradiated mice, as compared to NIR mice (Figure 6(Af)). These results suggest that 24 h post-IR, an early impairment of calcium homeostasis occurred, mainly in the sarcoplasmic reticulum of atrial cardiomyocytes (RyR2, SERCA2a, and PLN), associated with a defect in the maintenance of ionic gradients and, therefore, contractility, in atrial cardiomyocytes (NKA and Kir2.1).

Second, at a later stage, 60 weeks post-IR, our results showed a significant increase in the protein expression of RYR2 and SERCA2a in 0.5 Gy irradiated mice compared to NIR mice, only in the upper part of the heart (Figure 6(Ba,Bb)). The PLN pentamer/PLN monomer ratio was increased in 2 Gy irradiated mice only in the upper part of the heart (Figure 6(Bc)). The protein expression levels of NCX and NKA were not changed after irradiation in the upper and lower parts of the heart at 60 weeks post-IR compared to NIR mice (Figure 6(Bd,Be)). The Kir2.1 expression level was significantly increased in 0.5 Gy irradiated mice compared to NIR mice only in the upper part of the heart (Figure 6(Bf)). These results suggest that at 60 weeks post-IR, the previously observed early impairment of Ca^2+^ homeostasis in atrial cardiomyocytes (RYR2, SERCA2a, and PLN) was maintained and associated with a defect in atrial cardiomyocyte K+ gradients (Kir2.1).

In conclusion, our data showed that top-heart X-ray IR induced significant effects on the ECC at an early stage, mainly at 2 Gy, and that these effects persisted over time at the 0.5 Gy dose. These effects were mainly observed in atrial cardiomyocytes affected by localized irradiation, with an increase in the protein expression of the main sarcoplasmic reticulum actors in Ca^2+^ homeostasis, associated with a decrease in the expression of ion exchangers essential for the proper cardiomyocyte contraction.

## 3. Discussion

Our study showed that at low doses of ionizing X-rays (0.25 Gy) localized to the upper part of the heart, left ventricular diastolic dysfunction (LVDD) was impaired with a decrease in EF%, but preserved, and longitudinal deformation of the myocardium was altered. Short-term radiation-induced iatrogenic effects can be explained by the cytotoxic mode of action of ionizing radiation (apoptosis and disruption of Ca^2+^ homeostasis). In addition, an alteration in myocardial pump function was observed at 60 weeks post-IR. For the first time, our results showed a reorganization of the CSNS, observed by a significant early reduction in spines and dendrites at the branch level, as well as a late reorientation of the total number of nerve filaments in the heart. In addition, ionizing X-ray IR of the upper part of the heart led to a pro-fibrotic atrial remodeling 60 weeks post-IR and to early modulation of proteins involved in ECC pathways (Figure 7). However, the mechanisms involved in cardiac toxicity, particularly long-term arrhythmias, are more complex and, for these reasons, represent a public health problem that is still poorly understood.

These experimental findings of rhythm disturbances, diastolic dysfunction, EF% preservation, and strain alteration are consistent with the literature on patients treated with RT for BC. Indeed, Saiki et al. showed that the risk of heart failure (HF) or HF with preserved ejection fraction (HFpEF) increased with the increasing mean cardiac radiation dose received in a part of the heart [23]. Moreover, other studies showed that RT alone resulted in early reductions in the global and regional segmental diastolic strain rate over a follow-up period of 2 to 3 years [24,25]. In addition, the prevalence of AF showed that 2% to 16% of BC patients developed AF following chemotherapy, RT, or surgical ablation [26]. Although the risk of AF is more significant in patients over 65 years of age, patients with cancer have a 2-fold risk of developing stroke associated with AF and a 6-fold risk of developing myocardial infarction [13].

The accumulation of fibrosis observed in our study was mainly observed in the epicardial layer of the atria. It is well known that fibrosis leads to cardiac stiffening and conduction problems, which are associated with reduced ventricular function [27,28,29]. In our results, X-ray irradiation of the upper part of the heart increased caspase-3 expression (24 h and 60 weeks post-irradiation), leading to cell death in the atria, suggesting a loss of atrial function. Indeed, in certain contexts, atrial apoptosis and fibrosis were partly mediated by caspase-3, BAX, TGF-β1, and α-smooth-muscle actin [30]. Moreover, the burden on cardiomyocytes in the fibrotic heart was further increased, leading to additional cardiomyocyte death and replacement of lost cardiomyocytes by fibrotic deposits, resulting in a decline in cardiac function [27,31]. Thus, under our experimental conditions, we demonstrated a myocardial lesion induced by upper-heart X-ray IR, involving apoptosis (short and long term) of cardiomyocytes, atrial fibrosis, and remodeling, resulting in diastolic dysfunction and atrial conduction disturbances.

Furthermore, cardiac arrhythmias induced by ionizing radiation exposure are a long-term consequence characterized by cardiac electrical conduction, autonomic dysfunction, and cardiac tissue remodeling [17]. In our study, P-waves and AF susceptibility disappeared in all groups of irradiated mice. Indeed, one explanation could be long-term changes in CSNS, which could be the cause of our arrhythmia. We observed an increase in these catecholamines 24 h after IR at 2 Gy, and the disappearance of all the spines and dendritic arbors in the hearts of mice irradiated at 0.25 Gy. All this cardiac remodeling could be at the origin of the cardiac arrhythmias observed 60 weeks after IR at 2 Gy, as well as an increase and disappearance of all spines and dendritic arborizations in the hearts of mice irradiated at 0.25 Gy. Finally, a decrease in Sema3A was observed 60 weeks after irradiation, which was certainly related to the reorientation of nerve fibers within the heart rather than remaining at the periphery. It is well known that a wide range of cardiovascular effects (e.g., cardiac contractility and increased heart rate) depend on the CSNS, and dysfunction has led to pathophysiological processes in CD, such as heart failure and arrhythmia [32,33,34]. Cardiac sympathetic activity is mainly mediated by the neural catecholamines secreted (dopamine, noradrenaline, and adrenaline), which are important neurotransmitters and hormones responsible for positive inotropic effects (increased cardiac contractility), chronotropic effects (increased heart rate), lusitropic effects (acceleration of cardiac relaxation), and dromotropic effects (shortening of atrioventricular conduction) in the myocardium [34]. The arrhythmogenic substrate can be explained by an “irritable focus” resulting from myocardial fibrosis, elevated catecholamines, or a dysfunctional ion channel localized in the sarcolemma [35]. The CSNS also plays a key role in neurotransmitter-dependent regulation of cardiac nodes and myocardial activity. Indeed, the density of cardiac sympathetic nerves is maximal around the sinus node and increases progressively from the base of the ventricle to the apex [34]; in our case, the nerve fibers were internalized and, therefore, could not play their role optimally.

In addition, it is well established that low levels of catecholamines stimulate the heart by promoting Ca^2+^ movements, whereas excessive levels of catecholamines cause cardiac dysfunction by inducing intracellular Ca^2+^ overload in cardiomyocytes [36]. Furthermore, β1-adrenergic receptors are essential for autonomic regulation of the heart and are found in the SA and AV nodes, as well as in both atrial and ventricular cardiomyocytes. In addition, cardiac innervation is highly plastic and changes over time in different stages of CD [37,38]. They act by increasing the intracellular Ca^2+^ concentration, Ca^2+^ release from the SR, and by increasing the AVN conduction velocity [39]. Changes in intracellular Ca^2+^ concentration and cell size affect cell-to-cell coupling. Altered cellular coupling and fibrosis can alter anisotropic conduction, leading to further amplification of spatial non-uniformities and contractility troubles [40]. Electrophysiological disorders are characterized by abnormalities at the level of ion channels and modulation of the expression of trafficking proteins [41], and Marionneau et al. profiled 71 channels and associated genes in the sinoatrial node (SAN), atrioventricular node (AVN), atria (A), and ventricles (V) of adult male C57BL/6. These channels include the four structures RYR2, SERCA2a, NKA, and Kir2.1 [42]. In our study, we also observed an increase in the expression of these ion channels mainly in the irradiated part (atria and nodes) from 24 h post-IR, inducing a disruption of Ca^2+^ homeostasis, especially at 2 Gy. The major defects in Ca^2+^ cycling affect the sarcoplasmic reticulum (SR), the Ca^2+^ storage site in cardiomyocytes. Impairments in the regulation of Ca^2+^ cycling proteins, including the ryanodine receptor 2 (RyR2) and the sarcoplasmic/endoplasmic reticulum Ca^2+^ ATPase 2a (SERCA2a)/phospholamban (PLN) complex, are implicated in heart failure [43,44,45]. Moreover, intracellular Ca^2+^ concentrations are also influenced by small changes in the cytoplasmic Na^+^ content, thereby modulating cardiac contractility. The proteins involved in this Na^+^ homeostasis are the Na^+^/Ca^2+^ exchanger (NCX) and the Na^+^/K^+^ pump (NKA), which act to maintain Na^+^ influx mechanisms in cardiomyocytes [46]. Another component essential for the proper contractility of cardiomyocytes is the cardiac K^+^ inward rectifier channel, which is involved in cardiac repolarization and resting membrane stability [47,48]. Dysfunctional or overexpressed Kir2.1 channels disrupt the normal electrical activity of the heart, leading to irregular heart rhythms and potentially fatal arrhythmias. Kir2.1 channels mediate cellular electrical signaling, cellular communication, and their involvement in cardiovascular disease. Overexpression of Kir2.1 results in hyperpolarization of the resting membrane potential and shortening of the action potential [49,50]. Modulation of Kir2.1 expression, with gain and loss of function, has been implicated in the pathogenesis of several types of cardiac arrhythmias. Indeed, the two clinical phenotypes associated with a gain of Kir2.1 function are short QT syndrome type 3 (SQT3) and AF, both of which are extremely rare. Interestingly, patients with SQT3 are known to have both atrial and ventricular fibrillation, and the resting ECG shows features of extreme shortening of repolarization, with a short QT interval [51]. We observed this profile in our experiments (decreases in RR and QRS time).

Our results showed changes in cardiac function and ECG, reorganization of the CSNS at 60 weeks post-IR, and early modulation of proteins involved in ECC pathways, such as Kir2.1. These results are very promising for the identification of pro-arrhythmic mechanisms, the development of heart failure, and the future development of translational applications to prevent cardiotoxicity during RT treatment in BC patients. Indeed, considering new treatment strategies, such as (1) a step that would bypass certain areas of the heart that are particularly radiosensitive, such as the atria, the sinus-atrial node, and the atrioventricular node, or (2) new perspectives for the design of drugs just before RT, has allowed us to gain a better understanding of the pharmacology of ion channels, such as Kir2.1.

## 4. Materials and Methods

### 4.1. Animal Experimental Procedures

Animal experiments were performed in accordance with French and European regulations on the protection of animals used for scientific purposes (EC Directive 2010/63/EU and French Decree 2013–118). The entire study was approved by the Ethics Committee #81 of the Institute for Radiological Protection and Nuclear Safety (approval number E92-032-01) and authorized by the French Ministry of Research under the references APAFIS #31020-202104131246254 and APAFIS #40455-2023012315081234. All C57Bl/6J black mice were bred and maintained on a 12 h light/dark cycle with food and water ad libitum in a temperature-controlled room.

Two experimental time points were performed: an early time point 24 h after IR and a late time point 60 weeks after irradiation. At each time point, 8-week-old female mice were randomly divided into 4 groups of 12 mice each. As previously described, the cumulative doses received by the heart during RT for RBC ranged from 0.25 to just over 2 Gy [19]. Therefore, each group received SARRP (Small Animal Radiation Research Platform; xstrahl Ltd., Camberley, UK)-localized IR to the top of the heart at different doses: group 1 received 0.25 Gy, group 2 received 0.5 Gy, group 3 received 2 Gy, and group 4 represents the non-irradiated control (NIR) mice (Figure 8). At the end of each time point, the mice were randomly assigned according to the experiments to be performed: six animals per dose group for tissue analysis and six animals for molecular analysis. Euthanasia was performed using a ketamine–xylazine (100 mg/kg–10 mg/kg) mixture. The thorax was opened, and the heart was removed and weighed. The tibia was measured and used as a reference for heart weight. After removal, the hearts were immediately placed in saline solution to remove as much blood as possible from inside the organ. For histological experiments, hearts were fully embedded in OCT Tissutek and stored at −80 °C. For Western blot studies, hearts were divided into two parts: the upper part contained the atria, the cardiac nodes, and the top of the ventricles, and the lower part contained the ventricles. The samples were then immediately frozen at −80 °C until the experiment.

### 4.2. Irradiation Procedure

IR were performed on the SARRP (xstrahl Ltd., Camberley, UK) an image-guided preclinical irradiation platform that allows accurate targeted IR thanks to an onboard cone-beam computerized tomography (CBCT) system and dedicated treatment-planning MuriPlan software Version. 3.0.1 (Xstrahl, Inc., Suwanee, GA, USA) [52]. First, 8-week-old mice were anaesthetized with a ketamine/xylazine mixture. CBCT images were then acquired at 60 kV, 0.8 mA, with an inherent and additional filtration of 0.8 and 1 mm of Be and Al, respectively (2 min acquisition). These images were then used to plan the beam ballistics using the muri plan treatment-planning system [53]. A ~30% model of irradiation of the upper heart was developed (Appendix A), and mice were irradiated with a dose of 0.25, 0.5, or 2 Gy, with two beams at 180° to each other focused on an isocenter to obtain a homogeneous dose distribution. The heart was segmented, and the histogram dose volume was calculated to ensure that ~30% of the upper part was irradiated with a 3 × 9 mm collimator. Irradiations were performed using a voltage of 220 kV, an intensity of 3 mA, and an inherent and additional filtration of 0.8 and 0.15 mm of Be and copper, respectively, resulting in a half-value layer of 0.667 mm of copper. Control mice, hereafter referred to as non-irradiated (NIR) mice, were subjected to the same conditions as the irradiated mice, i.e., anesthesia and CBCT imaging, without SARRP irradiation.

### 4.3. Ultrasound Doppler and Speckle Tracking Analysis

Ultrasound Doppler studies were performed at the end of the protocol, 60 weeks post-IR, on 6 animals per dose group. High-resolution echocardiography was performed using the Vevo 3100 imaging system (system S/N:04N4Y8, version 3.2.7.15251; VisualSonics/Fujifilm, Toronto, ON, Canada) and the MX550D (25–55 MHz) transducer (VisualSonics, Fujifilm, Toronto, ON, Canada). Sedation was induced with 2% isoflurane and maintained at 0.5% isoflurane throughout the procedure, which averaged ≈6 to 10 min per mouse, during which the heart rate was maintained at 480–580 beats per minute. Mice were placed in a decubitus position on a heated platform at 37 °C to maintain body temperature. Electrocardiogram (ECG) and respiratory rate were monitored. The heart was imaged in 2D in the parasternal long-axis view (PSLAX) used to measure longitudinal strain, and M-mode short-axis images were used to measure several cardiac function parameters, including left ventricular ejection fraction (LVEF), left ventricular fractional shortening (LVFS), and diastolic left ventricular wall depth (LWd). Strain imaging analysis was performed using VevoStrain software in an offline workstation (Vevolab 3.8.1, VisualSonics/Fujifilm, Canada) using automatic tracing and speckle tracking echocardiography to detect the myocardial strain and strain rate. Global and regional longitudinal strain analysis was performed under the PSLAX. B-mode images of 300 frames at >200 frames/s were used for analysis, and PSLAX strain was calculated by tracking the movement of the endocardium and epicardium border in three consecutive cardiac cycles. PSLAX strain was calculated from the basal posterior, mid posterior, apical posterior, apical anterior, mid anterior, and basal anterior [54]. LV dyssynchrony was calculated as time-to-peak variation, defined as the standard deviation of time to peak over the six segments.

### 4.4. Atrial Burst Pacing

Atrial fibrillation (AF) episodes were induced by atrial burst pacing. Under anesthesia (ketamine–xylazine (100 mg/kg–10 mg/kg)), a 4-French stimulation catheter (Saint Jude medical, Boulogne-Billancourt, France) was inserted into the esophagus down to the level of the left atria. The left atria was then stimulated for 3 s at 10 Hz, and an ECG was recorded. AF duration was calculated as the time between the end of the stimulation and recovery of sinus rhythm [55].

### 4.5. Micro-CT Imaging

An in vivo micro-scanner platform was used for a longitudinal study performed before euthanasia. At 60 weeks post-IR, mice were anaesthetized under isoflurane inhalation (2%; 1 L/min O_2_) throughout the scanning process and i.v. injected with 100 µL of nanoparticle contrast agent, ExiTron™ nano 6000 (Viscover, Berlin, Germany). The micro-computed tomography (micro-CT) scans were performed on the Quantum GX2 (Perkin Elmer Health Sciences, Hopkinton, MA, USA) with cardiac gating, allowing images to be obtained in the diastolic and systolic phases. The images were acquired with a voltage of 90 kV, an intensity of 88 µA, an additional filtration of 0.06 mm of copper + 0.5 mm of aluminum, and a field of view of approximately 36 mm with a reconstruction of 25 mm, allowing a voxel size of 50 µm. The acquisition time was approximately 4 min, resulting in a mean absorbed dose estimated by the constructor to be approximately 0.9 Gy. Images were analyzed using the Analyze 15 software from AnalyzeDirect (AnalyzePro Windows Installation, AnalyzeDirect, Overland Park, KS, USA), with semi-automatic and manual segmentation (volume quantification) of the left and right atria in systole. The same threshold range was used for all mice to allow comparison. 

### 4.6. Immunohistological Assessment of Heart Injury

Here, 10 µm longitudinal sections of hearts were cut using a cryostat (CryoStar NX50, Epredia, Portsmouth, NH, USA) and serially mounted on glass slides for histological analysis. Fibrosis was revealed with the Picrosirius Red Stain Kit (Connective Tissue Stain, ab150681; Abcam, Cambridge, UK). After staining, all sections were imaged using a whole slide scanner, NanoZoomer S60 (#C13210-01, Hamamatsu Photonics, France), at a 40X objective using NZAcquire software (version 3.1.10). The fibrosis quantification was analyzed using Histolab^®^ automated image analysis software (version 12.2.2, Microvision Instruments, Lisses, France). For immunolabeling of H2AX phosphorylation at Ser 139 (γ-H2AX; Abcam; ref ab11174), the 1:500 dilution was used at 4 °C overnight after blocking with 5% BSA for 1 h. After washing 4 times for 5 min, Goat anti-Rabbit IgG (H + L) Highly Cross-Adsorbed Secondary Antibody, Alexa Fluor™ 546 (Cat # A-11035; Thermo Fisher Scientific, Waltham, MA, USA), was incubated at 1/300 dilution.

### 4.7. Whole-Mount Immunolabeling Using Tissue Clearing Method (iDISCO+)

Hearts were fixed overnight at 4 °C by immersion in 4% paraformaldehyde (PFA) before starting the iDISCO+ protocol. The iDISCO whole-mount immunolabeling protocol was based on the work of Renier and collaborators [56]. We performed this technique on whole hearts from all experimental groups at 24 h and 60 weeks post-IR. Quantifications were based on the schematic representation of the algorithms for automated tracking of sympathetic nervous system filaments immuno-stained by TH with the semi-automated microscope image analysis software IMARIS (version x64 10.1.0; Oxford Instruments, Belfast, Northern Ireland, UK; Appendix A).

Briefly, fixed samples were washed three times for 30 min in phosphate-buffered saline (PBS), then dehydration in increasing concentrations of methanol (in H_2_O) was carried out (20–40–60–80–100–100%), with each step lasting 1 h. The samples were then incubated overnight in 66% dichloromethane (DCM)/33% methanol. After two 30 min washes in 100% methanol, samples were bleached with 5% H_2_O_2_ in methanol (1 vol. 30% H_2_O_2_/5 vol methanol, ice cold) overnight at 4 °C. After bleaching, samples were rehydrated in decreasing concentrations of methanol (80–60–40–20%) and, finally, in PBS for 1 h. Then, samples were incubated twice for 1 h in PBS/0.2% Triton X-100 and then permeabilized in PBS/0.2% Triton X-100/20% DMSO/60.3 M glycine at 37 °C overnight. Samples were blocked in PBS/0.2% Triton X-100/10% DMSO/6% donkey serum at 37 °C for two days. Samples were incubated at 37 °C for 10 days with the primary antibody tyrosine hydroxylase (TH; AB152, dilution 1/200; Sigma-Aldrich, Saint-Louis, MI, USA) to reveal dopaminergic neurons, specifically from the sympathetic nervous system, and diluted in PBS/0.2% Tween20 with 10 µg/mL of heparin (PTwH)/5% DMSO/3% donkey serum. Samples were then washed five times for 30 min in PTwH and incubated with the secondary antibody donkey anti-rabbit Alexa647 (A32795, dilution 1/400; Thermo Fisher Scientific, Waltham, MA, USA) diluted in PTwH/3% donkey serum at 37 °C for four days. Then, samples were washed five times for 30 min in PTwH before clearing and imaging. Samples were incubated in 66% dichloromethane (DCM)/33% methanol for 3 h, followed by two 15 min incubations in 100% DCM. Finally, samples were incubated in dibenzyl ether (DBE) until clear and then stored in DBE for at least 24 h before light-sheet microscopy. Samples were incubated in ethylcinnamate (ECI) to perform acquisition (Ultramicroscope II, LaVision Miltenyi Biotec, CEA, Fontenay-aux-Roses). The Ultramicroscope II is monitored by the software Imspector Pro (version 10, LaVision BioTec License 64bit image acquisition- and processing software for Microsoft Windows) and equipped with an Andor Néo sCMOS camera (Andor Technologies, Belfast, Irland), a 2X Olympus MVLAPO lens, and a variable magnification from 0.63 to 6.3X Olympus MVX-10 magnification body for Ultramicroscope (Olympus, Tokyo, Japan). Samples were excited with a laser diode at 488 nm (488-85 Laser for LVBT Laser Module Gen. II), with laser transmission set at 50% in a bidirectional manner. The sheet width was set at 80% to illuminate the entire sample, and the thickness was set at 4.1 to maximize the z-resolution. Emitted fluorescence was detected using a 525/50 nm band-pass filter. Light-sheet fluorescent microscopy enabled 3D reconstructions of the imaged samples by stacking all the images taken step-by-step. This microscope was used at the 1X magnification. Z-stack images were acquired with a suitable step size (2 µm) along the normal direction to encompass the entire sample volume and acquire either a coronal or sagittal view. Due to the very high dynamic range (15 bit) of the light-sheet images, a gamma correction and a median filter were applied to the light-sheet datasets for display purposes. IMARIS software was used for 3D manipulations using automated filament tracing algorithms (Appendix A).

### 4.8. Western Blot Analysis

Heart lysates from frozen heart tissue, previously divided into two parts before freezing, were prepared using radioimmunoprecipitation assay (RIPA) lysis buffer containing a protease/phosphatase inhibitor cocktail (#04693116001, Roche, Bâle, Suisse). Protein concentrations in the lysates were determined using BCA reagent. Equal amounts (40 µg) of proteins in the lysates were separated on a 12-well 4–15% SDS-polyacrylamide gel electrophoresis (SDS-PAGE) gel (Mini-PROTEAN^®^ TGX™ Precast Gel, #4561085; Bio-Rad, Hercules, CA, USA) and transferred to a polyvinylidene difluoride (PVDF) membrane (Trans-Blot Turbo Mini 0.2 µm PVDF Transfer, #1704156; Bio-Rad, Hercules, CA, USA). The membrane was incubated in 5% skim milk in PBS containing 0.1% Tween 20 to block non-specific binding. The blot was probed with appropriate primary antibodies against Sema3A (PA5-67972, dilution 1/1000; Thermo Fisher Scientific, Waltham, MA, USA), cleaved caspase-3 (9661, dilution 1/1000; Cell Signaling Technologies, Danvers, MA, USA), SERCA2a (SAB5701015, dilution 1/750; Sigma-Aldrich, Saint-Louis, MI, USA), PLN (1/1000; PA5-85268, Thermo Fisher Scientific, Waltham, MA, USA), RYR2 (MA3-916, dilution 1/1000; Thermo Fisher Scientific, Waltham, MA, USA), NaK-ATPase (SAB5701012, dilution 1/1000; Sigma-Aldrich, Saint-Louis, MI, USA), Kir2.1 (PA5-16635, dilution 1/1000; Thermo Fisher Scientific, Waltham, MA, USA), β1-adrenergic receptor (ADRB1, PA5-28808, dilution 1/1500; Thermo Fisher Scientific, Waltham, MA, USA), and GAPDH (ab8245, dilution 1/20.000; Abcam, Cambridge, UK) overnight at 4 °C. Immunoreactive protein bands were incubated with HRP-conjugated anti-mouse or rabbit antibodies for 1 h at room temperature and detected with ECL reagents (Substrate Chemiluminescent SuperSignal™, #34578; Thermo Fisher Scientific, Waltham, MA, USA). Membranes were imaged using an Amersham ImageQuant ™ 800 Western blot imaging system (Cytiva Life Science, Marlborough, MA, USA) and quantified using ImageQuant™ TL 10.0 analysis software (Cytiva Life Science, Marlborough, MA, USA). Protein expression was normalized to GAPDH. Results are from two independent experiments.

### 4.9. ELISA Catecholamines

For the quantitative determination of mouse catecholamine concentrations in heart tissue homogenate, a catecholamine ELISA kit was used according to the manufacturer’s instructions (#ABIN772979; https://www.antibodies-online.com (accessed on 28 August 2024)). The samples were prepared by ribolysis, and a BCA protein assay (Pierce Rapid Gold BCA Protein Assay Kit, A53225; Thermo Scientific Scientific, Waltham, MA, USA) was performed in parallel to determine the protein concentrations in the samples. Absorbance was read at 450 nm (Infinite 200 PRO Microplate Reader, #1412003467; Tecan, Männedorf, Suisse).

### 4.10. Ca^2+^ Assay

A Ca^2+^ assay kit (ab102505, Abcam, Cambridge, UK) was used to measure Ca^2+^ concentrations in heart tissue homogenate according to the manufacturer’s instructions. Samples were prepared by ribolysis, and a BCA protein assay (Pierce Rapid Gold BCA Protein Assay Kit, A53225; Thermo Scientific Scientific, Waltham, MA, USA) was performed in parallel to determine the protein concentrations in the samples. Absorbance was read at 575 nm (Infinite 200 PRO Microplate Reader, #1412003467; Tecan, Männedorf, Suisse).

### 4.11. Statistical Analysis

Results are expressed as mean ± SEM and compare the experimental groups with NIR mice. Based on the results obtained with the Shapiro–Wilk normality test for samples with *n* < 50, the following tests were applied: for samples following a normal distribution, an ordinary one-way ANOVA test with Dunnett’s multiple comparison post hoc test was used; for samples not following a normal distribution, a Kruskal–Wallis test with Dunn’s multiple comparison post hoc test was used. The *p*-values of <0.05 were considered significant. Analyses were performed with GraphPad Prism 8 software.

## Figures and Tables

**Figure 1 ijms-25-09483-f001:**
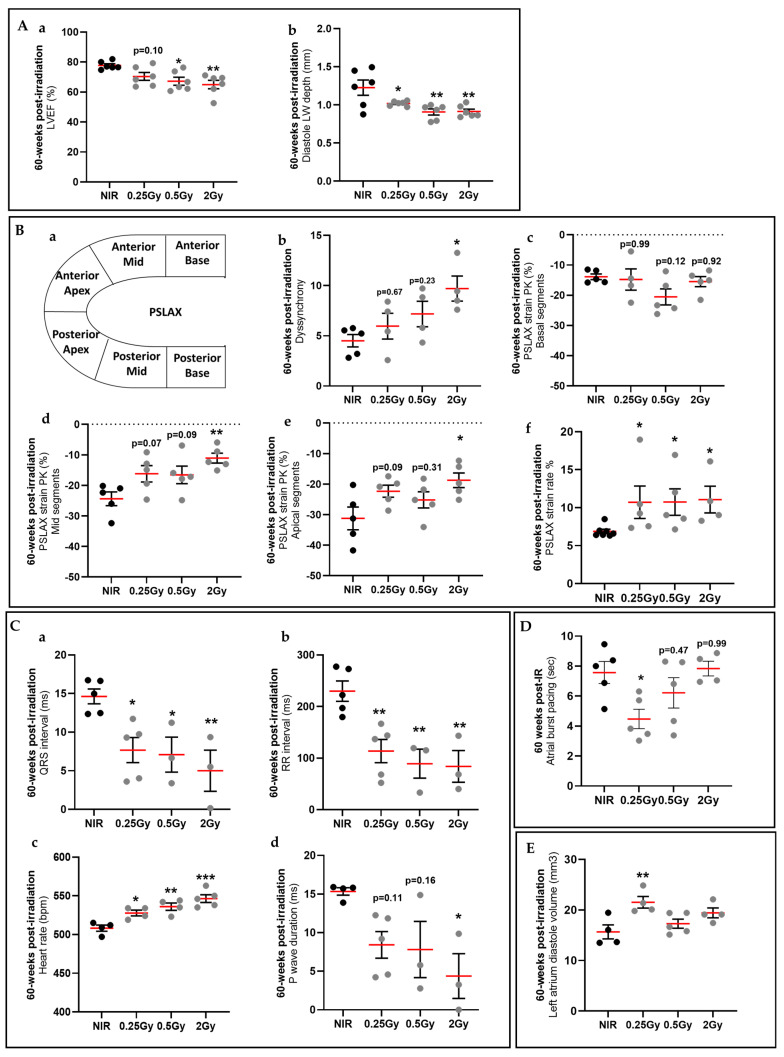
Impact of 30% top-heart X-ray irradiation on atrial and ventricular cardiac functions in mice 60 weeks after irradiation. (**A**) Ventricular cardiac functions were measured by echocardiography in M-mode: left ventricular ejection fraction (LVEF) (**a**) and diastole left ventricular wall depth (LW) (**b**). Data were presented as mean ± SEM (*n* = 5–6 per group). * *p* < 0.05, ** *p* < 0.01 vs. NIR mice by one-way ANOVA test followed by Dunnett’s multiple comparison post hoc test. (**B**) Schematic representation of specific myocardial regions and segments used for measurement of myocardial deformation by myocardial strain analysis (**a**). Measurement of left ventricular (**b**) dyssynchrony, (**c**–**e**) longitudinal deformation in specific myocardial regions, and (**f**) strain rate. Data were presented as mean ± SEM (*n* = 4–5 per group). * *p* < 0.05 and ** *p* < 0.01 vs. NIR mice by one-way ANOVA test followed by Dunnett’s multiple comparison post hoc test or Kruskal–Wallis test followed by Dunn’s multiple comparison post hoc test. (**C**) Electrocardiographic (ECG) analysis with summary of QRS interval (**a**), RR interval (**b**), heart rate (**c**), and P-wave duration (**d**). Data were presented as mean ± SEM (*n* = 3–5 per group). * *p* < 0.05, ** *p* < 0.01, and *** *p* < 0.005 vs. NIR mice by one-way ANOVA test followed by Dunnett’s multiple comparison post hoc test or Kruskal–Wallis test followed by Dunn’s multiple comparison post hoc test. (**D**) Post-pacing pause recorded following transesophageal atrial fibrillation. Left atrium diastole volume (mm^3^). Data were presented as mean ± SEM (*n* = 4–5 per group). * *p* < 0.05 vs. NIR mice by one-way ANOVA test followed by Dunnett’s multiple comparison post hoc test. (**E**) Micro-CT analysis of left atrium diastole volume 60 weeks post-IR. Data were presented as mean ± SEM (*n* = 4–6 per group). ** *p* < 0.01 vs. NIR mice by one-way ANOVA test followed by Dunnett’s multiple comparison post hoc test.

**Figure 2 ijms-25-09483-f002:**
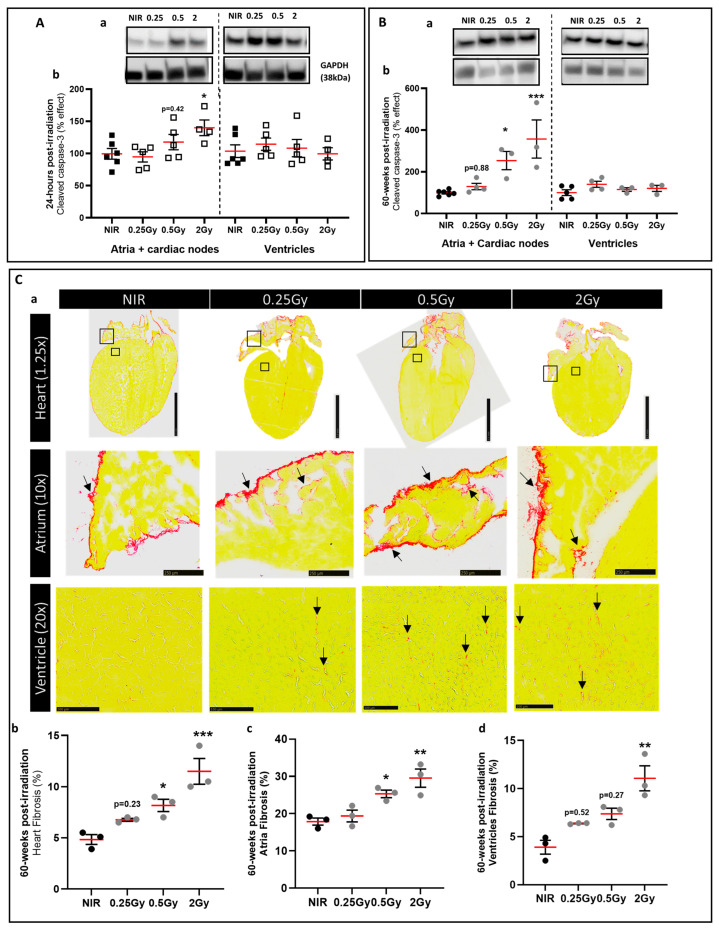
The 30% top-heart X-ray irradiation induced fibrosis and apoptosis mainly in atria. (**A**) Western blot analysis of the cleaved caspase-3 protein in atria and ventricles in heart of mice 24 h post-IR (**a**). GAPDH was used as an internal control. Data were shown as a percentage effect ± SEM compared to NIR mice (*n* = 4–6 per group from 2 independent experiments) (**b**). * *p* < 0.05 vs. NIR mice by one-way ANOVA test followed by Dunnett’s multiple comparison post hoc test. (**B**) Western blot analysis of the cleaved caspase-3 protein in atria and ventricles in heart of mice 60 weeks post-IR. GAPDH was used as an internal control (**a**). Data were shown as a percentage effect ± SEM compared to NIR mice (*n* = 3–5 per group from 2 independent experiments) (**b**). * *p* < 0.05 and *** *p* < 0.005 vs. NIR mice by one-way ANOVA test followed by Dunnett’s multiple comparison post hoc test. (**C**) Representative images demonstrating patterns of interstitial fibrosis by Picrosirius Red staining (collagen fibers stained red) on longitudinal cardiac tissue sections in the heart, atria, and ventricles from all experimental groups at 60 weeks post-IR (scale bar: 2.5 mm, 250 µM, and 100 µM, respectively) (**a**). Black squares indicate the location of the zoomed region of interest in the atria and ventricles. Arrows indicate interstitial fibrosis. Quantification of cardiac fibrosis is presented on the right. Data were presented as mean ± SEM (*n* = 3 per group) (**b**–**d**). * *p* < 0.05, ** *p* < 0.01 and *** *p* < 0.005 vs. NIR mice by one-way ANOVA test followed by Dunnett’s multiple comparison post hoc test.

**Figure 3 ijms-25-09483-f003:**
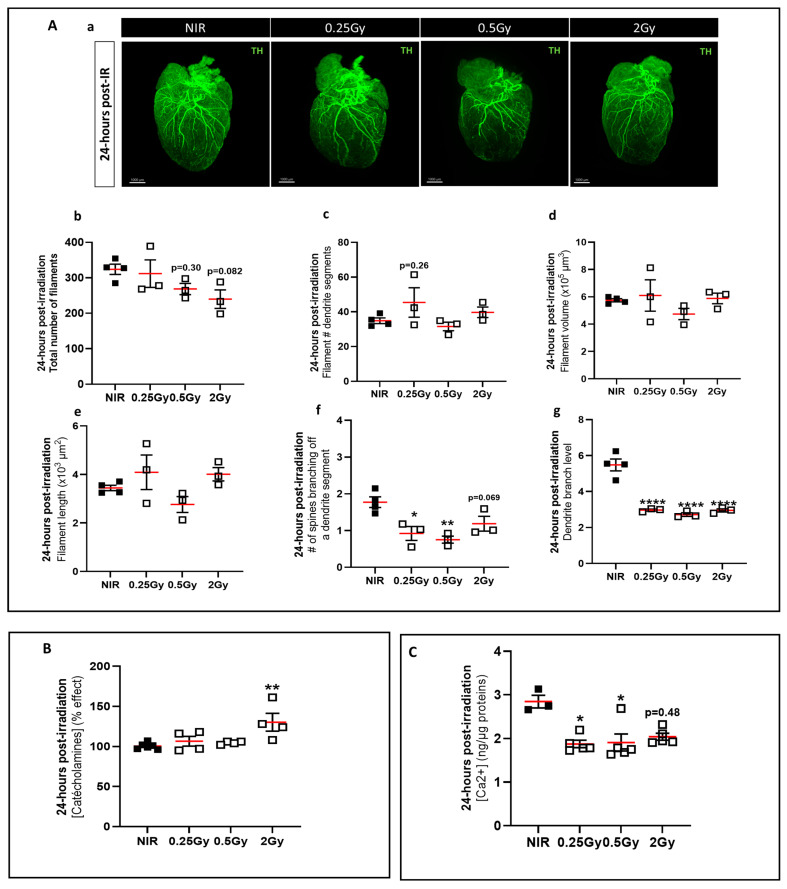
Early impact of 30% top-heart X-ray irradiation on cardiac SNS morphology. (**A**) Images of iDISCO+ cleared hearts of mice 24 h post-IR acquired by light-sheet fluorescence microscopy (**a**). Quantification of TH-positive fibers in the whole heart using automated filament tracer in IMARIS software in mice 24 h after irradiation: total number of filaments (**b**), filament volume (**c**), filament length (**d**), number of dendritic segments on filaments (**e**), spine density on a dendritic segment (**f**), and branch level (**g**). Data were presented as mean ± SEM (*n* = 3–4 per group). Scale bar: 1000 µm. * *p* < 0.05, ** *p* < 0.01, **** *p* < 0.001 vs. NIR mice by one-way ANOVA test followed by Dunnett’s multiple comparison post hoc test. (**B**) Cardiac catecholamines levels in mice 24 h post-IR. Catecholamines were measured by ELISA in heart samples. Data were shown as a percentage effect ± SEM compared to NIR mice (*n* = 4–5 per group). ** *p* < 0.01 vs. NIR mice by one-way ANOVA test followed by Dunnett’s multiple comparison post hoc test. (**C**) Cardiac calcium concentration in mice 24 h post-IR. Calcium concentrations were measured by a colorimetric assay in heart samples. Data were shown as a percentage effect ± SEM compared to NIR mice (*n* = 3–5 per group). * *p* < 0.05 vs. NIR mice by Kruskal–Wallis test followed by Dunn’s multiple comparison post hoc test.

**Figure 4 ijms-25-09483-f004:**
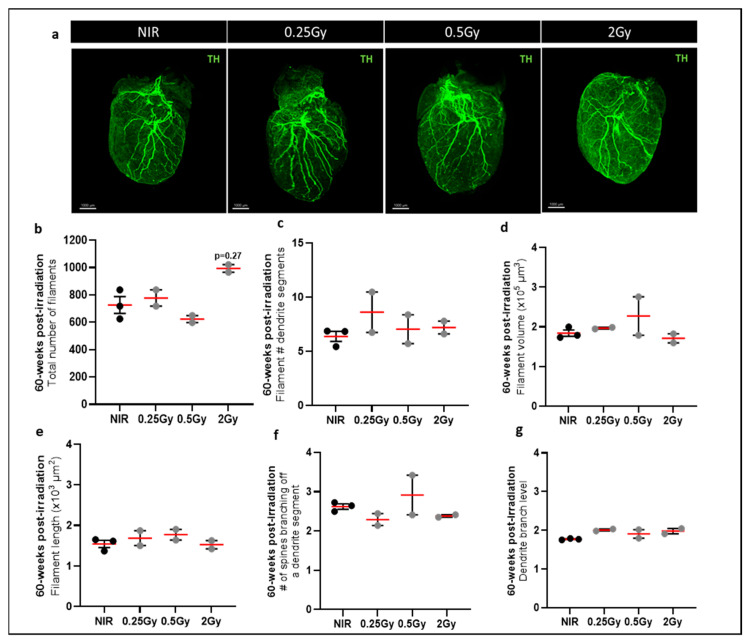
General IMARIS analysis of the cardiac sympathetic nervous system 60 weeks post-irradiation. Images of iDISCO+ cleared hearts of mice 60 weeks post-IR acquired by light-sheet fluorescent microscopy (**a**). Quantification of TH-positive fibers in the whole heart using the automated filament tracer in IMARIS software in mice 24 h after irradiation: total number of filaments (**b**), filament volume (**c**), filament length (**d**), number of dendritic segments on filaments (**e**), spine density on a dendritic segment (**f**), and branch level (**g**). Data were presented as mean ± SEM (*n* = 2–3 per group). Scale bar: 1000 µm.

**Figure 5 ijms-25-09483-f005:**
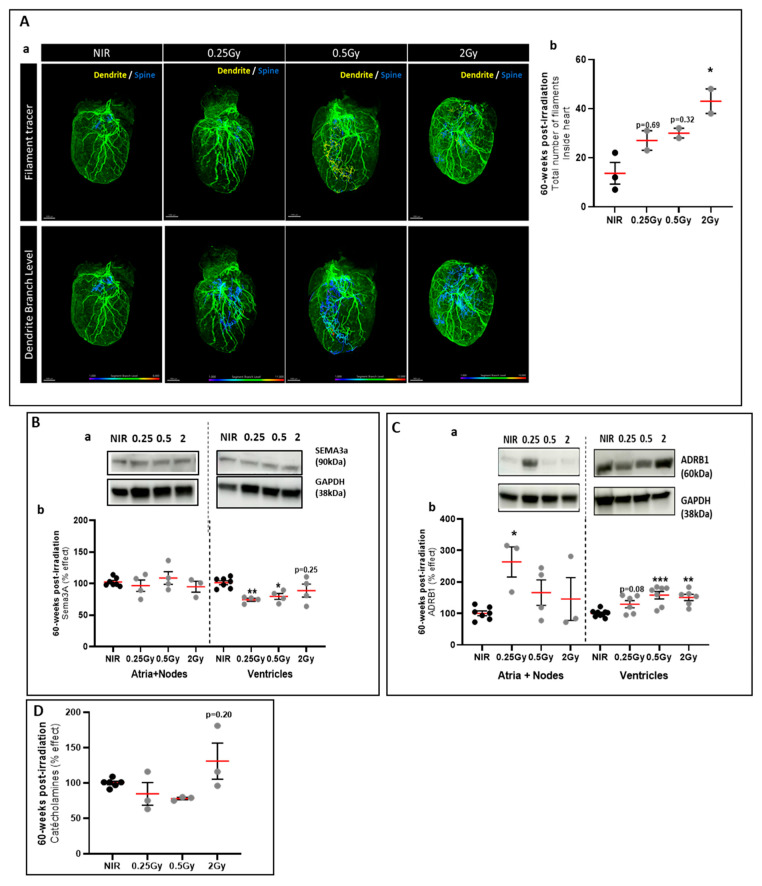
The 30% top-heart X-ray irradiation induced a delayed reorganization of the cardiac SNS. (**A**) IMARIS analysis images (**a**) and quantification of TH-positive fibers inside the heart (**b**) using an automated filament tracer in IMARIS software in mice 60 weeks post-IR. Data were presented as mean ± SEM (*n* = 2–3 per group). * *p* < 0.05 vs. NIR mice by Kruskal–Wallis test followed by Dunn’s multiple comparison post hoc test. (**B**) Western blot analysis of the Sema3a in hearts of mice 60 weeks post-IR (**a**). GAPDH was used as an internal control. Data were shown as a percentage effect ± SEM compared to NIR mice (*n* = 3–6 per group from 2 independent experiments) (**b**). * *p* < 0.05 and ** *p* < 0.01 vs. NIR mice by one-way ANOVA test followed by Dunnett’s multiple comparison post hoc test. (**C**) Western blot analysis of ADRB1 protein in hearts of mice 60 weeks post-IR (**a**). GAPDH was used as an internal control. Data were shown as a percentage effect ± SEM compared to NIR mice (*n* = 3–7 per group from 2 independent experiments) (**b**). * *p* < 0.05, ** *p* < 0.01 and *** *p* < 0.005 vs. NIR mice by one-way ANOVA test followed by Dunnett’s multiple comparison post hoc test. (**D**) Cardiac catecholamine levels in mice 60 weeks post-IR. Catecholamines were measured by ELISA in heart samples and data were shown as a percentage effect ± SEM compared to NIR mice (*n* = 3–4 per group).

**Figure 6 ijms-25-09483-f006:**
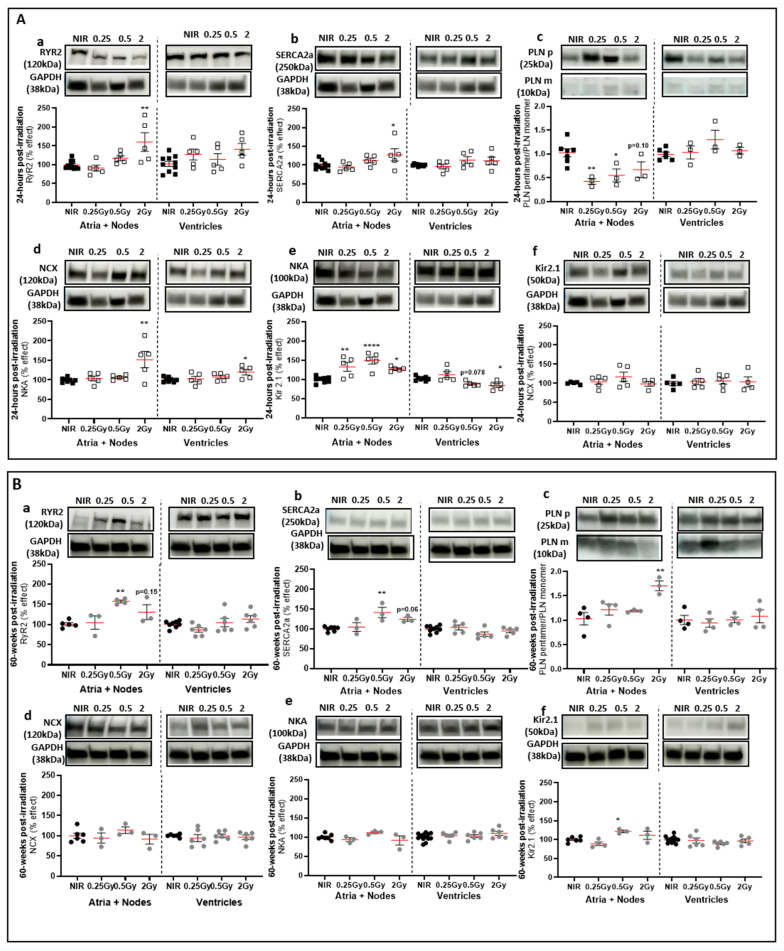
Effect of 30% top-heart X-ray irradiation on calcium homeostasis. (**A**) Western blot analysis of the calcium proteins in atria and ventricles of mice 24 h post-IR. RYR2 (**a**), SERCA2a (**b**), PLNp/PLNm ratio (**c**), NCX (**d**), NKA (**e**) and Kir2.1 (**f**). GAPDH was used as an internal control. Data were shown as a percentage effect ± SEM compared to NIR mice (*n* = 5–8 per group from 2 independent experiments). * *p* < 0.05 and ** *p* < 0.01, **** *p* < 0.001 vs. NIR mice by one-way ANOVA test followed by Dunnett’s multiple comparison post hoc test. (**B**) Western blot analysis of the calcium proteins in atria and ventricles of mice 60 weeks after irradiation. RYR2 (**a**), SERCA2a (**b**), PLNp/PLNm ratio (**c**), NCX (**d**), NKA (**e**) and Kir2.1 (**f**). GAPDH was used as an internal control. Data were shown as a percentage effect ± SEM compared to NIR mice (*n* = 3–6 per group from 2 independent experiments). * *p* < 0.05 and ** *p* < 0.01 vs. NIR mice by one-way ANOVA test followed by Dunnett’s multiple comparison post hoc test.

**Figure 7 ijms-25-09483-f007:**
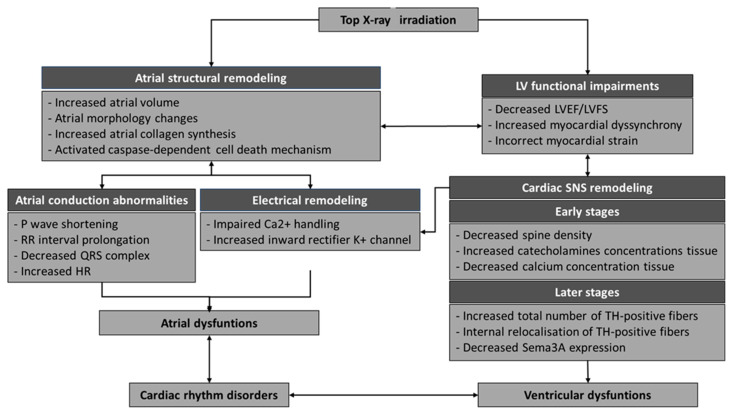
Schematic summary of the impact of ionizing radiation on the top of the heart.

**Figure 8 ijms-25-09483-f008:**
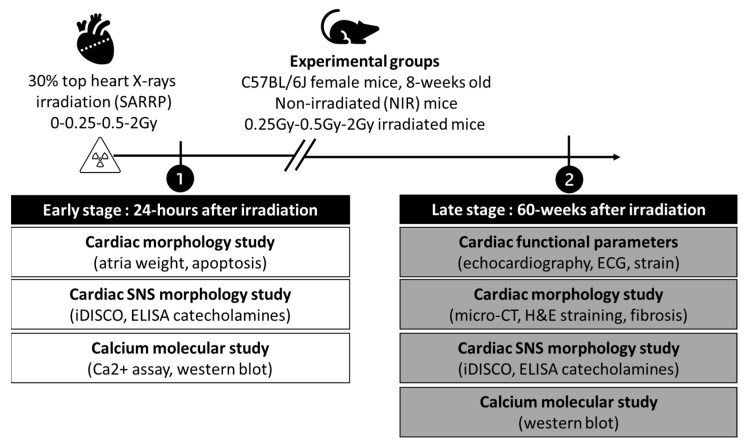
Study design and localized irradiation procedure. The study design was divided into three major time points, namely, localized irradiation of the hear with SARRP, the early time point 24 h after irradiation, and the late time point 60 weeks after irradiation. Female C57BL/6J mice were divided into 4 experimental groups: non-irradiated control (NIR) mice, and 0.25, 0.5, and 2 Gy irradiated mice. At each time point, hearts were collected. Different experiments were then carried out, such as cardiac function, cardiac morphology, CSNS morphology, and cardiac molecular analysis.

## Data Availability

The raw data supporting the conclusions of this article will be made available by the authors upon request.

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
