# Peer review of "X-ray Radiotherapy Impacts Cardiac Dysfunction by Modulating the Sympathetic Nervous System and Calcium Transients"

_ijms, 2024, doi:10.3390/ijms25179483_

Round 1

Reviewer 1 Report

Comments and Suggestions for Authors

Feat-Vetel et al. investigated the acute (24 h) and late chronic (60 weeks) effects of irradiation (0.25-2 Gy) focused on the upper part of the heart on the cardiac sympathetic nervous system, atrial remodeling, and calcium homeostasis in female mice in their descriptive study. Their major findings are that i) cardiac and atrial dysfunction developed with atrial fibrosis and apoptosis at week 60; ii) expression of proteins related to calcium homeostasis (e.g., Ryr2, NAK, Kir2.1 and Serca2a) increased in the atrium in a dose-dependent manner and persisted over time; and iii) early reduction in spines and dendrites of the cardiac sympathetic nervous system and a late reorientation of nervous fibers with a decreased expression of SEMA3a were detectable in their radiotherapy model. The topic is interesting and novel, and the experiments seem to have been carefully designed and performed using plenty of methods. This reviewer has only minor comments.

1. The authors claim in the methods section that 6 animals were used in each group. In contrast, the sample number is only 2-3 in several experiments in some groups (e.g., Western blots, ECG analysis). Why did the authors not use all of the samples? 

2. Figure 7: it is hard to read the axis descriptions. Please increase the fonts.

3. In the Supplemental material, please indicate which band belongs to which group on the blots.

4. Beyond the strains, did you measure other conventional diastolic dysfunction parameters (e.g., e', E/e')?

5. You might write a limitations section on the lower sample number in several experiments and the potential effects of aging, cardiovascular risk factors, and sex-based differences on your results.

Comments on the Quality of English Language

1. Please remove or correct the fragment sentences in the text (e.g., page 2, line 66: "The cardiac sympathetic nervous system (CSNS) and the AV/SA.", page 9, line 329: "Using the VevoStrain software for LV deformation analysis.")

2. Several sentences are hard to follow (e.g., page 2, lines 54-56: Indeed, Bansod et al. show in a non-trivial 2.3% of breast cancer (BC) survivors hospitalized who received RT during 55 treatment developed an arrhythmia death over 7 years").

3. Please insert a space between the numbers and units (e.g., 0.25Gy vs. 0.25 Gy) and use the upper index in the cases of ions (e.g., Ca2+ vs. Ca2+).

4. Introduce the abbreviations at their first mention (e.g., CBCT in line 127 vs. line 128).

5. Please try to simplify the long sentences and check the text with English grammar software for clarity and correctness also.

Author Response

Reviewer 1

Comments and Suggestions for Authors

Feat-Vetel et al. investigated the acute (24 h) and late chronic (60 weeks) effects of irradiation (0.25-2 Gy) focused on the upper part of the heart on the cardiac sympathetic nervous system, atrial remodeling, and calcium homeostasis in female mice in their descriptive study. Their major findings are that i) cardiac and atrial dysfunction developed with atrial fibrosis and apoptosis at week 60; ii) expression of proteins related to calcium homeostasis (e.g., Ryr2, NAK, Kir2.1 and Serca2a) increased in the atrium in a dose-dependent manner and persisted over time; and iii) early reduction in spines and dendrites of the cardiac sympathetic nervous system and a late reorientation of nervous fibers with a decreased expression of SEMA3a were detectable in their radiotherapy model. The topic is interesting and novel, and the experiments seem to have been carefully designed and performed using plenty of methods. This reviewer has only minor comments.

First, We would thank the reviewer 1 for the interest in our work

Comments 1. The authors claim in the methods section that 6 animals were used in each group. In contrast, the sample number is only 2-3 in several experiments in some groups (e.g., Western blots, ECG analysis). Why did the authors not use all of the samples? 

Answer 1: Thank you for pointing that out. In response to your comment, I would like to explain the difference in the number of animals used. In total, there are twelve irradiated animals per dose group. In the functional study, six animals were used (VEVO, VEVO strain, atrial burst pacing, micro-CT, ECG). However, the number of animals varied between three and six because, depending on the experiment, some animals might give unusable results (individual heterogeneity in the response of some of the magnets) or die during the experiment. The twelve irradiated animals were then randomly divided into two groups: a group of six animals for the histological study (fibrosis (3 animals) and iDISCO+(3 animals)) and a group of 6 animals for the molecular study (WB). I have therefore modified the manuscript in more detail:

  • Page 3 line 102 and 108-110 paragraph: “At each time point, 8-week-old female mice were randomly divided into four groups with twelve mice each. […] At the end of each time point, mice were randomly assigned according to the experiments to be performed: six animals per dose group for tissue analysis and six animals for molecular analysis.”
  • Page 4 line 142: “Ultrasound Doppler studies were performed at the end of the protocol, 60 weeks post-IR, on six animals per dose group.”

Comments 2: Figure 7: it is hard to read the axis descriptions. Please increase the fonts.

Answer 2: I agree with you. I have changed the fonts in all the graphs on Figure 7 accordingly (page 17).

Comments 3: In the Supplemental material, please indicate which band belongs to which group on the blots.

Answer 3: I agree with you. I have notified the bands on Supplemental material accordingly.

Comments 4: Beyond the strains, did you measure other conventional diastolic dysfunction parameters (e.g., e', E/e')?

Answer 4: We did not measure other conventional parameters of diastolic dysfunction. All the parameters quantified are listed in the table S1 in Supplemental material.

Comments 5: You might write a limitations section on the lower sample number in several experiments and the potential effects of aging, cardiovascular risk factors, and sex-based differences on your results.

Answer 5: Thank you for your comment. The small size of the samples obtained at the end of in vivo studies is a limitation due either to individual heterogeneity (for functional experiments), or to the death of the oldest animal during the experiment, giving unusable results. However, given the cost and duration of our innovative experiments (specialty in iDisco), we wanted to start with a small number of animals but sufficient to be representative and significant. We are currently relaunching experiments that should confirm and extend these results with a larger scale study. Regarding the possible effects of age and cardiovascular risk factors, we conducted a long-term study (60 weeks post-irradiation) in our experimental conditions to be as close as possible to what happens in women after radiation treatment for breast cancer. However, it would be useful to supplement our results with a subsequent dose of radiation therapy, for example 52 weeks instead of 8 weeks, and to study the long-term effects after radiation therapy.  Gender difference: The context of our study is the effects of X-ray irradiation in radiation therapy for breast cancer. This is why our study model is based on female mice only and therefore we did not study the difference between the sexes. In particular, conduction abnormalities and arrhythmias were the most common clinical manifestations observed in post-RI patients. Bansod et al show in a non-trivial study that 2.3% of hospitalized breast cancer (BC) survivors who received radiation therapy during treatment experienced arrhythmia death over 7 years [8]Interestingly, patients, who did not receive RT during their treatment, did not develop arrhythmias or conduction disturbances [9], and the risk of developing these rhythm disturbances is higher in RBC patients [7, 10].” (page 2 lines 53-59)

Comments on the Quality of English Language

Comments 1: Please remove or correct the fragment sentences in the text (e.g., page 2, line 66: "The cardiac sympathetic nervous system (CSNS) and the AV/SA.", page 9, line 329: "Using the VevoStrain software for LV deformation analysis.")

Answer 1: Thank you for pointing this out. I have removed the line 66, and corrected page 9 line 331 to read  “Using the VevoStrain software for LV deformation analysis, we showed a significant increase in dyssynchrony in 2Gy irradiated mice as compared to NIR mice”.

Comments 2: Several sentences are hard to follow (e.g., page 2, lines 54-56: Indeed, Bansod et al. show in a non-trivial 2.3% of breast cancer (BC) survivors hospitalized who received RT during 55 treatment developed an arrhythmia death over 7 years").

Answer 2: Thank you for your comments. I have changed all the sentences that were too long and difficult to understand.

Comments 3: Please insert a space between the numbers and units (e.g., 0.25Gy vs. 0.25 Gy) and use the upper index in the cases of ions (e.g., Ca2+ vs. Ca2+).

Answer 3 : Thank you for your comments. Throughout the manuscript I have added a space between numbers and units and the top index in the case of ions.

Comments 4. Introduce the abbreviations at their first mention (e.g., CBCT in line 127 vs. line 128).

Answer 4: Thank you for pointing this out. I have introduced the abbreviations at their first mention.

Comments 5: Please try to simplify the long sentences and check the text with English grammar software for clarity and correctness also.

Answer 5: As replied in comment 2, I have changed all the sentences that were too long and difficult to understand.

Reviewer 2 Report

Comments and Suggestions for Authors

The Authors focused their study on the recent epidemiological studies that have shown patients with right breast cancer (RBC)  treated by X-ray (IR) irradiation, are more susceptible to develop cardiovascular diseases such as arrhythmias, atrial fibrillation and conduction disorders after radiotherapy (RT). The objective of the study was  to examine the mechanisms induced by low to moderate doses of IR and to evaluate alterations in  the cardiac sympathetic nervous system (CSNS), atrial remodeling and calcium homeostasis involved in the heart rhythm. The Authors   mimic the RT of the RBC in  female C57Bl/6J mice with  X-ray doses ranging from 0.25 to 2Gy targeting 40% of the top of the heart. The results showed  that from 0.25Gy, local heart irradiation induced late cardiac and atrial dysfunction and fibrosis  development. After irradiation, CSNS in the ventricle and the calcium transient protein expression were rearranged, which had an impact on cardiac contractility. Results are very promising in terms of identifying pro-arrhythmic mechanisms and to prevent arrhythmias during RT treatment in RBC patients.

Thanks for giving me the opportunity to review this interesting study conducted with scientific rigor. However, I would like to ask the authors of the minority issues:

 ·     Could induce death with euthanasia affect the data of the analysis of post-         death cardiac tissues?

  ·    Was the Dose delivered to the mice with the Cone Beam CBCT evaluated?        However, it is in addition to the dose ranges delivered for the study (0.25-          2Gy - control mice).

My opinion is, Minor Revision.

Comments on the Quality of English Language

The Authors focused their study on the recent epidemiological studies that have shown patients with right breast cancer (RBC)  treated by X-ray (IR) irradiation, are more susceptible to develop cardiovascular diseases such as arrhythmias, atrial fibrillation and conduction disorders after radiotherapy (RT). The objective of the study was  to examine the mechanisms induced by low to moderate doses of IR and to evaluate alterations in  the cardiac sympathetic nervous system (CSNS), atrial remodeling and calcium homeostasis involved in the heart rhythm. The Authors   mimic the RT of the RBC in  female C57Bl/6J mice with  X-ray doses ranging from 0.25 to 2Gy targeting 40% of the top of the heart. The results showed  that from 0.25Gy, local heart irradiation induced late cardiac and atrial dysfunction and fibrosis  development. After irradiation, CSNS in the ventricle and the calcium transient protein expression were rearranged, which had an impact on cardiac contractility. Results are very promising in terms of identifying pro-arrhythmic mechanisms and to prevent arrhythmias during RT treatment in RBC patients.

Thanks for giving me the opportunity to review this interesting study conducted with scientific rigor. However, I would like to ask the authors of the minority issues:

 ·     Could induce death with euthanasia affect the data of the analysis of post-         death cardiac tissues?

  ·    Was the Dose delivered to the mice with the Cone Beam CBCT evaluated?        However, it is in addition to the dose ranges delivered for the study (0.25-          2Gy - control mice).

My opinion is, Minor Revision.

Author Response

Reviewer 2

Comments and Suggestions for Authors

The Authors focused their study on the recent epidemiological studies that have shown patients with right breast cancer (RBC)  treated by X-ray (IR) irradiation, are more susceptible to develop cardiovascular diseases such as arrhythmias, atrial fibrillation and conduction disorders after radiotherapy (RT). The objective of the study was  to examine the mechanisms induced by low to moderate doses of IR and to evaluate alterations in  the cardiac sympathetic nervous system (CSNS), atrial remodeling and calcium homeostasis involved in the heart rhythm. The Authors   mimic the RT of the RBC in female C57Bl/6J mice with X-ray doses ranging from 0.25 to 2Gy targeting 40% of the top of the heart. The results showed that from 0.25Gy, local heart irradiation induced late cardiac and atrial dysfunction and fibrosis development. After irradiation, CSNS in the ventricle and the calcium transient protein expression were rearranged, which had an impact on cardiac contractility. Results are very promising in terms of identifying pro-arrhythmic mechanisms and to prevent arrhythmias during RT treatment in RBC patients.

Thanks for giving me the opportunity to review this interesting study conducted with scientific rigor. However, I would like to ask the authors of the minority issues:

My opinion is, Minor Revision.

First, we thank Reviewer 2 for his interest in our work and the time he has devoted to revising the manuscript. 

Comments 1: Could induce death with euthanasia affect the data of the analysis of post-death cardiac tissues?

Answer 1: Thank you for bringing this to our attention. Euthanasia is performed by cervical dislocation (no cardiac involvement) and to avoid damage to the cardiac tissue after euthanasia, samples are directly collected and frozen at -80°C , as described in the manuscript (page 3 lines 120-125): After extraction, The hearts were immediately placed in saline to remove as much blood as possible from inside the organ. For the histological experiments, the hearts were fully integrated into the Tissutek OCT and stored at - 80°C. For western-blot studies, the hearts were divided into two parts: the upper part contained the atria, the heart nodes and the lower part contained the ventricles. The samples were then immediately frozen at -80°C until the experiment.”

Comments 2: Was the Dose delivered to the mice with the Cone Beam CBCT evaluated?  However, it is in addition to the dose ranges delivered for the study (0.25-2Gy - control mice).

Answer 2: Thank you for your comment. As indicated in the text on page 4, lines 130, for CBCT, the acquisition time is approximately 2 minutes, giving an average absorbed dose estimated by the manufacturer to be less than 5 cGy (0,05Gy). This dose is relatively low and negligible. However, to ensure that the control mice are under the same experimental conditions as the irradiated mice, ‘control mice, hereafter referred to as non-irradiated (NIR) mice, were subjected to the same conditions as the irradiated mice, i.e. anesthesia and CBCT imaging, without SARRP irradiation’ (page 4, lines 139-140).